

# Social big data management through collaborative mobile, regional, and cloud computing

Afzal Badshah[1], Ameen Banjar[2], Safa Habibullah[2], Abdullah Alharbi[3], Wael Alosaimi[3] and Ali Daud[4]

[1] Department of Software Engineering, University of Sargodha, Sargodha, Punjab, Pakistan
[2] Department of Information Systems and Technology, College of Computer Science and Engineering, University of Jeddah, Jeddah, Saudi Arabia
[3] Department of Information Technology, College of Computers and Information Technology, Taif University, Taif, Saudi Arabia
[4] Faculty of Resilience, Rabdan Academy, Abu Dhabi, United Arab Emirates

## ABSTRACT

The crowd of smart devices surrounds us all the time. These devices popularize social media platforms (SMP), connecting billions of users. The enhanced functionalities of smart devices generate big data that overutilizes the mainstream network, degrading performance and increasing the overall cost, compromising time-sensitive services. Research indicates that about 75% of connections come from local areas, and their workload does not need to be migrated to remote servers in real-time. Collaboration among mobile edge computing (MEC), regional computing (RC), and cloud computing (CC) can effectively fill these gaps. Therefore, we propose a collaborative structure of mobile, regional, and cloud computing to address the issues arising from social big data (SBD). In this model, it may be easily accessed from the nearest device or server rather than downloading a file from the cloud server. Furthermore, instead of transferring each file to the cloud servers during peak hours, they are initially stored on a regional level and subsequently uploaded to the cloud servers during off-peak hours. The outcomes affirm that this approach significantly reduces the impact of substantial SBD on the performance of mainstream and social network platforms, specifically in terms of delay, response time, and cost.

# INTRODUCTION

Smartphones have high-resolution cameras and sensors, allowing them to record high definition (HD) video, audio, and photos. Similarly, the social media platform (SMP) enables users to upload or upstream these high-resolution audios and videos to the mainstream network (*Xu et al., 2020*; *Badshah et al., 2022b*). Furthermore, the emerging Artificial Intelligence (AI) is integrated with these platforms and devices, indicating that social media users (along with the AI devices) can increase the overall population as the network-connected devices outnumber the overall population. The statistics show that more than 75% of the data on the internet belongs to social networks and streaming

Corresponding author
Afzal Badshah,
afzalbadshahkhattak@gmail.com

applications (*Statista, 2021*; *Rao & Saraswathi, 2021*). Therefore, the burning issue that arises from the massive crowd of devices and SMP is the social big data, which overloads the mainstream network, decreases the SMP performance, and increases the cost (*Hung, Wang & Hwang, 2020*; *Tran et al., 2017*).

Analysis shows that about 4.28 billion people were using mobile phones in 2020. It further states that 90% of internet users use the internet from mobile. The mobile forecast seems to increase rapidly with the affordable new trends and technology (*Appel et al., 2020*). These devices are used to connect to SMP. Statistics show that 3.78 billion people are connected to social websites, which is anticipated to reach 4.41 billion by 2025 (*Statista, 2021*).

According to survey findings, 75% of social media friends come from the same country where the users reside. It means most data must not be routed to the cloud server. Therefore, we have proposed a collaborative structure of mobile edge computing (MEC), regional computing (RC) (*Badshah et al., 2022a*), and cloud computing (CC) to filter data at the cellular and regional level to reduce the workload on the primary network during peak hours. However, all content is transferred to the cloud servers during off-peak hours due to the non-availability of scalable capacity on regional servers or mobile edge. Figure 1 shows the collaborative working of MEC, RC, and CC. Table 1 shows the acronyms used in the article.

To initially evaluate this hypothesis, the Microsoft Azure online tool is used to measure the delay, response time, and cost among different areas around the globe (*Azure Speed Test, 2021*). The upload and delay time around the region (as shown in Table 2) for 100 kB per request is calculated. The findings indicate that these parameters increase with the growing distance between the server and end-users (*Azure Speed Test, 2021*).

Therefore, this study aims to handle this massive workload locally using mobile and regional computing instead of putting it on the mainstream network's shoulders. The earlier version of this article used regional computing to handle social big data regionally (*Badshah et al., 2022a*). Mobile, regional, and cloud computing collaboration will filter the content at the mobile and regional levels before transferring it to the cloud. It will result in a lower load on mainstream networks, high performance, and lower cost of the SMP.

Given the above challenges, the main objectives of this study are:

- Enable efficient content search on nearby smart devices and regional servers (with user permission) to reduce delay and dependence on remote cloud servers.
- Alleviate the burden of SMP-generated workloads on mainstream networks by implementing regional-level content filtering during peak hours and scheduling cloud server uploads during off-peak hours.
- Improve application performance and reduce costs by offloading workloads from SMP cloud servers through regional-level filtering and processing.

Edge and fog computing paradigms are positioned close to users, processing and storing data at the edge rather than transmitting it to the cloud (*Badshah et al., 2022a*; *Liu et al., 2024*). Edge computing dramatically decreases processing time, transfer duration, and

**Figure 1** Working of mobile edge computing (MEC), regional computing (RC) and cloud computing (CC).

**Table 1 Acronym used in this study.**

| Acronym | Explanation |
|---|---|
| CC | Cloud Computing |
| RC | Regional Computing |
| EC | Edge Computing |
| MEC | Mobile Edge Computing |
| FC | Fog Computing |
| MC | Mobile Computing |
| BS | Base Station |
| SMP | Social Media Platforms |
| IoT | Internet of Things |
| AI | Artificial Intelligence |
| HD | High Definition |
| PT | Packet Tracer |

**Table 2 Upload time and delay among different regions of the world.**

| Server | End users | Load | Upload time (sec) | Delay (ms) |
|---|---|---|---|---|
| Pakistan (Asia) | Singapore (Southeast Asia) | 100 kB | 2.37 s | 297 ms |
| Pakistan (Asia) | Johannesburg (South Africa) | 100 kB | 2.55 s | 422 ms |
| Pakistan (Asia) | Ireland (Europe) | 100 kB | 3.53 s | 365 ms |
| Pakistan (Asia) | Toronto (Canada) | 100 kB | 4.21 s | 600 ms |
| Pakistan (Asia) | Lowa (Central America) | 100 kB | 3.02 s | 500 ms |

expenses (*Dautov et al., 2018*; *Goudarzi et al., 2021*). However, they are not well-equipped to handle big data, especially the data generated by social media. Due to the long-distance location of their cloud servers, delays and high workloads result in low performance and

increased costs (*Abughazalah et al., 2024*). This issue can be addressed by mobile and regional computing, which has already attracted approximately $104 billion in global investments (*Statista, 2021*; *Badshah et al., 2020a*).

The proposed idea may be explained clearly with a daily life scenario:

*On social media users' requests for content, the RC algorithm checks its availability in the user's base station (BS) smart devices. If the content is available on any device, it is streamed to the requested device through the base station, not affecting the regional or cloud servers. Similarly, to overcome the worst effects of the workload generated by SMP on the consumer network, regionally filter content during peak hours and adjust downloads to cloud servers during off-peak hours.*

Continuing the preceding discussion, the primary contributions of this article are:

- Collaboration of MEC, RC, and CC by using their computational and storage for regional data processing and storing to minimize the overburden on the mainstream network.
- Search the content in nearby smart devices and regional servers instead of searching and transferring data from distanced social cloud servers.
- Improving the performance of the social media platform (SMP) by utilizing algorithms on edge and regional computing servers to determine the optimal placement of workload (*i.e.*, MEC, RC, and CC).

The organization of the remaining study is outlined as follows: "Literature" presents a comprehensive literature review, offering a contextual backdrop. Subsequently, "Proposed Model" articulates the methodology employed to address the challenges associated with the impact of social big data. Moving forward, "Evaluation" critically evaluates the outcomes of experimentation conducted through simulations. "Open Issues" investigates the lingering open issues and challenges within the study, providing insight into areas that warrant further exploration. "Conclusion", the key findings are succinctly summarized, and future directions for research are outlined.

## LITERATURE

Research indicates that around 3.29 billion users will be connected to social media by the end of 2022 (*Statista, 2021*). It represents around 43% of the world's population. Moreover, according to the CISCO report, it is projected that by 2023, 66% of the population will have internet connectivity, contributing to the continuous growth of social media. Millions of users visit these platforms daily and spend hours there (*CISCO, 2021a*). This population creates massive amounts of data that overloads the main internet stream. This section covers related work to minimize social media load on the conventional network and maximize SMP performance.

Leading internet firms such as SpaceX and Google are actively working to reduce round-trip delays for big data. SpaceX, in particular, has revolutionized the approach by delivering internet connectivity directly from space, utilizing satellites in lower orbit rather

than relying on traditional cables and cellular networks (*StarLink, 2021*). The difference between Starlink (SpaceX) and other geostationary satellites is that the geostationary satellites are 35,000 km from Earth, which has a significant delay. However, the Starlink satellites orbit 550 km away, providing a fast internet connection. They have already started to market internet connections. However, this technology is still in its initial stages and faces several challenges and criticisms.

The perspective on (SMP) is broad, as it is widely acknowledged that social media encompasses various elements such as people, content, information, behavior, and organizations within an interconnected environment that supports interactivity (*Appel et al., 2020*). As an example, Facebook, as of March 31, 2019, reported 2.38 billion monthly active users and 1.56 billion active users (*Statista, 2021*). Additionally, it is projected that the global total of social media users will reach 4.41 billion by 2025, constituting 60% of the world's population (*Statista, 2021*). To cover the depth, the literature is categorized into edge computing, mobile computing, and hybrid environment, considering delay minimization.

## Edge computing

Edge or fog computing is a recent world terminology aimed at extending cloud services to the network's edge to reduce delay, response time, and costs. Edge computing (EC) is now massively used, especially in uncrewed vehicles and games (*Ndikumana et al., 2021*). *Gu & Zhou (2019)* proposed an algorithm, An Approximate Load Balancing Task Offloading Algorithm (ALBOA), to distribute vehicle workloads to different edge servers for minimum processing time. The experimental result shows that the algorithm achieves its objectives. Similarly, *Chen et al. (2018)* investigated resource-efficient edge issues for smart IoT applications. The proposed algorithm divides the smart application workload into similar nearby devices to minimize the delay. Their evaluation report stated that the proposed algorithm minimizes the delay of the service. *Niu et al. (2019)* explored the minimum service delay in Internet of Things (IoT) edge computing. Their algorithm divides the allocation into three different steps. First, the load balancing is carried out. Secondly, resources are allocated to the workload, and finally, converting the problem into an algorithm to solve it.

EC is useful in delay, response, and cost minimization. However, this domain needs to be expanded to MEC after the crowd of smart devices and technological shifts, making these devices practical for MEC.

## Mobile computing

There is a crowd of smart devices around us. *Fernando, Loke & Rahayu (2019)* focused on this crowd and used nearby mobile devices to share the workload burden and not migrate it to edge or cloud computing. It saves energy and minimizes the cost and delay. Mobile devices use Bluetooth and WiFi to connect for data sharing. *Fernando, Loke & Rahayu (2019)* proposed a honeybee mobile computing structure using Bluetooth and WiFi for data sharing instead of cellular networks. The main challenge of the proposed structure lies in the limited battery power of Android devices. In the case of mobile computing, mainly

when devices communicate directly, mobile collaboration is required, allowing mobile devices to respond to queries. The authors study the related concept in *Mtibaa et al. (2013)* to keep mobile energy consumption at a minimum, specifically in case the team is working in an emergency. Their proposed framework determines where to perform the task so that minimum energy can be consumed.

Direct sharing among mobile devices is an interesting idea; however, it has two main barriers. The first is the battery life, and the second is the users' consent. Although the battery life is advancing, it still needs to be in a position to support continuous communication. Similarly, the standard operating procedures (SOPs) of direct communication need to be developed.

## Hybrid environment

The collaborative structure of edge and cloud is usually used for delay, response, and cost minimization. *Loghin, Ramapantulu & Teo (2019)* explored the hybrid environment of cloud and edge computing and calculated the processing and transfer time on edge servers, cloud servers, and hybrid. Their comparison result shows that edge computing decreases the processing and transfer time.

Statistics show that about 46% of internet traffic is generated by mobile devices. To address these issues, *Chen, He & Qiao (2019)* proposed a mobile edge computing concept at the university level. The Universities' infrastructure is used as servers, and the students' mobiles use these resources.

The IoT devices produce big data. Large servers are required at the central processing unit to process the incoming data. *Wan et al. (2019)* used an uncrewed aerial vehicle-based station (UAV-BS), hovering at the sensors. The UAV-BS has sufficient capacity to process sensor data locally; however, when necessary, this data is transferred to the cloud server.

The most significant challenge in the cloud is communication delay. Despite efforts to minimize delay through job offloading, certain limitations persist. To tackle these challenges, the authors in *Azure Speed Test (2021)* suggested a collaboration between edge computing and cloud computing aimed at improving performance and lowering power consumption.

Likewise, in *Ren et al. (2019)*, the authors suggested a collaboration between cloud and edge computing to tackle these challenges. They utilized the cloud for computing services and aimed at minimizing edge latency. Their findings indicate that the proposed approach effectively reduces latency in cloud communications. The evaluation results underscore that the collaboration of edge and cloud surpasses the individual use of either cloud or edge computing.

Mitigating the data overload from multimedia applications, a collaborative structure of mobile edge and cloud computing was proposed (*Wu et al., 2017*). To improve the user experience, they implemented strategies for device-to-device and machine-to-machine communication, effectively reducing the burden on base stations. Simulation results demonstrated a significant decrease in the workload on base stations.

A 30-frame video, having a resolution of 1024 × 1024, requires 125 TB of storage space when played 100 times per hour. Due to its high capacity requirements,

*Kioumourtzis et al. (2017)* conducted research indicating that closed circuit television (CCTV) video streaming poses a risk of system-wide failure. In response, they introduced a hybrid cloud and edge computing concept, incorporating various video encoding techniques to takle this challenge. Experimental results demonstrated that the MPEG4 encoding scheme yielded a more reliable performance.

Operating in a hybrid environment poses several challenges. Firstly, specialized algorithms and technology are required to switch tasks between the edge and the cloud efficiently. Secondly, the necessity of transferring workloads to mobile computing has arisen with recent smart technology and the proliferation of smart devices. Therefore, this aspect requires further exploration.

## PROPOSED MODEL

The proposed idea, depicted in Fig. 2, functions across four layers: (i) Smart devices layer, (ii) Cellular communication layer, (iii) Regional computing layer, and (iv) Social cloud computing layer.

The social media's cloud layer incorporates servers with enhanced storage and processing capacity. Initially, regional data is stored on edge servers and later transferred to the social media cloud server during off-peak hours. MEC uses the BS to share heavy files among mobile devices instead of downloading from cloud servers. The communication infrastructure connects the mobile devices, and the smart devices share the content through this layer. Table 3 illustrates the notation utilized in this section.

### Devices layer

This layer covers all devices connected to the communication system and the delay and cost caused by the data transfer from these devices to the base station, regional server, and cloud server.

Before explaining the working of the layer mentioned above, it is worth reporting here to explain the delays the communication faces (*Bagubali et al., 2018*). Four types of delay accumulate in communication. Transmission delay ($D_{tran}$) is to transmit the data to the transmission media, propagation delay ($D_{prop}$) is the data travel time on the transmission media from source to destination, queuing delay ($D_q$) is the data waiting for the time in a queue to be processed by the system, and processing delay ($D_{proc}$) when the system processes the data.

To compute the transmission delay ($D_{tran}$),

$$D_{\text{tran}} = \frac{W \times SNR \times ME}{B \times \text{ER}} \tag{1}$$

where $W$ is the workload (data), $B$ is the channel bandwidth capacity, $SNR$ is the signal-to-noise ratio (*Zaman et al., 2023*), modulation efficiency ($ME$) is the efficiency of the modulation scheme, and $ER$ is the rate of transmission errors (*Zaman et al., 2022*).

The equation shows that the data transmission speed will also be high if the channel has a high bandwidth.

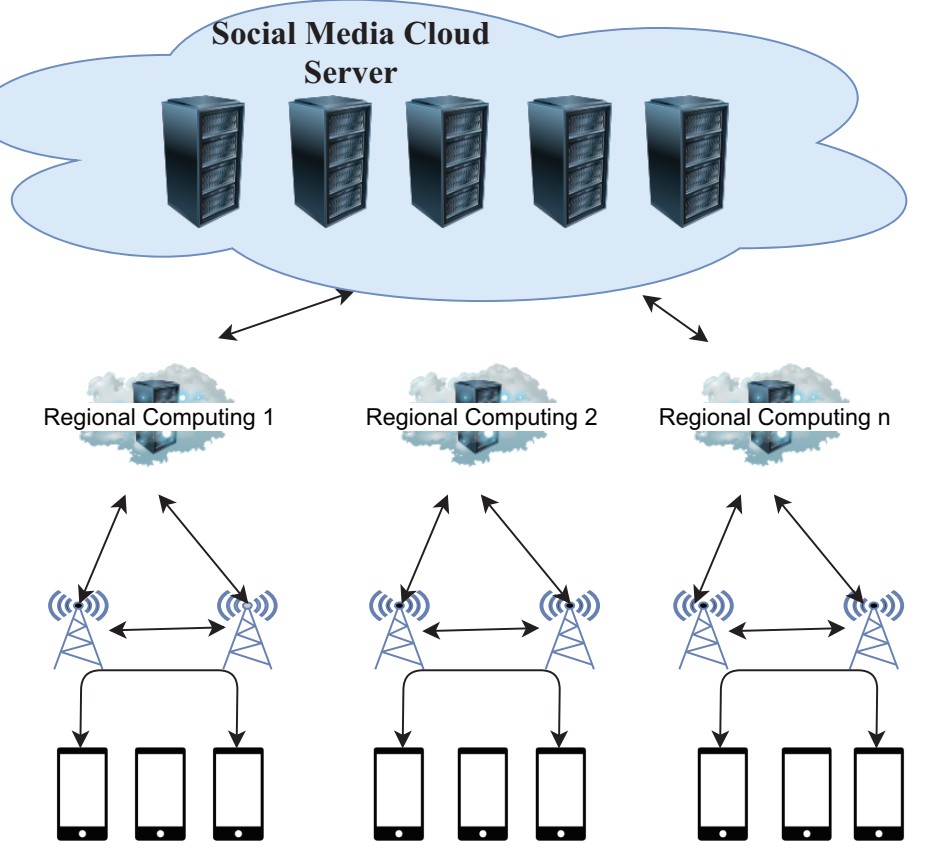

Cloud Computing work as a centralized server to store and process all the social media data, located at a wide distance from the end-users.

Regional Computing server store and process the regional data, not needed outside the region. However, this data is uploaded to cloud in low traffic time or in case of high popularity.

Cellular communication layer, have the base station and cellular towers provides platform to connect mobile, regional and cloud computing.

The smart devices layer covers all the devices connected to social media. This creates big data in the form of text, photos, audios and videos.

**Figure 2 Proposed collaboration structure of mobile computing, regional computing, and cloud computing.**

**Table 3 Symbols and notation employed in the article.**

| Symbol | Representation | Symbol | Representation |
|---|---|---|---|
| $D_{tran}$ | Transmission delay | $D_{prop}$ | Propagation delay |
| $D_{que}$ | Queuing delay | $D_{proce}$ | Processing delay |
| $D_t$ | Total delay | $W$ | Workload |
| $T_s$ | Channel capacity | $Dis$ | Distance |
| $tr_s$ | Transmission speed | $D_{com}$ | Communication delay |
| $DM_t$ | Total delay faced while interacting to cloud servers | $D_{me}$ | Delay between mobile device and edge |
| $D_{er}$ | Delay between edge and regional servers | $D_{rc}$ | Delay between regional and cloud computing servers |
| $E_{total}$ | Total energy utilization on data transmission | $E_{tran}$ | shows the total energy utilization on data transfer |
| $E_{stor}$ | Shows the total energy consumption in storage | $E_{col}$ | Shows the total power consumption to cool the data centers |
| $Cost_{operational}$ | Operational cost | | |

Similarly, to calculate the propagation delay ($D_{prop}$),

$$D_{prop} = \frac{Dis}{tr_s} \qquad (2)$$

where $Dis$ represents the distance, and $tr_s$ denotes the transmission speed.

To calculate the processing delay ($D_{proc}$),

$$D_{proc} = \frac{S}{P_r} \qquad (3)$$

where $S$ is the size of the data and $P_{pr}$ is the processing rate of the particular machine.

Likewise, when the arrival of data increases the transmission or processing capacity, the data has to wait in the queue. The queuing delay ($D_q$) depends on congestion.

The queuing delay is calculated as

$$D_q = \frac{L \times a}{R} \qquad (4)$$

where $L$ is the packet length, $a$ is the arrival rate of packets, and $R$ is the packet processing rate.

The delay in mobile communication also depends on the devices connected to the base station; therefore,

$$D_{MCI} = \frac{BSC}{NMC} \qquad (5)$$

$D_{MC}$ is the delay caused by mobile connections impact, $BSC$ is a base station capacity and $NMC$ is the number of connected devices.

Therefore, the total delay faced by the data is calculated as;

$$D_t = D_{tran} + D_{prop} + D_{proc} + D_{que} + D_{MCI} \qquad (6)$$

where $D_t$ shows the total delay data faces to reach the destination. $D_{tran}$ is the transmission delay, referring to the time from the transfer of the first bit to the last bit to the transmission medium. $D_{prop}$ shows the propagation delay; as the name refers, it shows the delay faced when the data propagates through the network. Similarly, $D_{proc}$ shows the data processing delay, and $D_{que}$ shows the queuing delay. For this study, the transmission delay ($D_{tran}$) and propagation delay ($D_{prop}$) are the key factors affecting the overall delay.

The layers discussed in this study use different channels and have different distances between users and servers. So, the delay will change according to the scenario.

## Cellular communication layer

This article introduces this layer to share large files (*i.e.*, images and videos) through BS station instead of streaming data from cloud servers and regional servers. It not only minimizes the delay and response time, but it also minimizes the operational cost.

The mobile devices delay is computed as follows:

$$DM_t = 2 \times D_{tran} + 2 \times D_{prop} + D_q + D_{proc} \qquad (7)$$

$DM_t$ shows the total delay of interaction with the base station, while $D_{tran}$ represents the transmission delay for transmitting the data. Similarly, $D_{prop}$ is the propagation delay between the mobile and base station, $D_q$ is the queuing delay, and $D_{Proc}$ is the processing delay at the base station. Propagation delay is minimum at MEC however, we have

included it due to the wireless delay. In the case of mobile communication with BS, the queuing delay ($D_q$) needs to be addressed because there is no masses connection with BS.

Algorithm 1 receives a content request from the users, and the optimum content location is suggested to stream the data to the target device with minimum time and cost. The base station keeps a list of the smart devices connected to it. The content is searched on these devices' shared content. If the search is successful, the data is streamed to the target device; in the event of failure, the request is forwarded to the regional server.

The experimental cellular communication delay for ping is 12 ms. According to the research, this must stay the same by 20 ms.

The smart device layer encompasses all devices connected to the social network, including computers, laptops, smartphones, tablets, AI devices, and isolated devices. The communication layer consists of mobile networks that connect smart devices to edge computing. The anticipated 5G and 6G technologies provide the capability to transfer data instantly and without delay between devices and edge servers.

### Regional layer

The Regional Computing server serves as the backbone of the proposed structure. It employs algorithms and AI technology to determine which content to store on the Edge Server or transfer to the cloud. The content is filtered using Algorithm 1. Content deemed unnecessary in another region is neither transferred to the cloud nor processed and stored locally. However, this data is transferred to the cloud servers during off-peak hours.

The delay faced by the regional server is calculated as follows:

$$DR_t = D_{me} + D_{er} \tag{8}$$

$DR_t$ is the total delay faced while interacting with regional servers, and $D_{me}$ is the delay between mobile device and edge. Similarly, $D_{er}$ is the delay between edge and regional servers.

Similarly, the complexity of the regional servers depends on the total capacity of the regional server and the total workload on it.

$$\text{RC} = \frac{CL}{MCC} \tag{9}$$

$RC$ is the regional computing complexity, $CL$ is the computational load, and $MCC$ is the maximum computational capacity.

Algorithm 1 receives a content request from the users, and the optimum content location is suggested to stream the data to the target device with minimum time and cost. This algorithm receives the search request on the failed search at the base station. The content is searched on the regional servers. Upon a successful search, the data is streamed to the target devices, and in case of failure, the request is forwarded to the cloud servers.

### Cloud layer

Cloud computing is the umbrella layer in this system, responsible for storing all data. However, the majority of computing services are distributed across the regional servers to

---

**Algorithm 1 Algorithm for MEC, regional computing, and cloud search.**

**Input:** Request for content (identifier), shared list of mobile devices, address table of regional servers and attached base stations, user permissions, and network performance metrics.

**Output:** Location of content (mobile, base station, regional server, or cloud), successful delivery to the user, or an error message if unavailable.

When the content request is received:

Populate the mobile devices shared list and address table of regional servers and attached base stations

**if** *content is found in nearby mobile devices (MEC)* **then**

    Transfer the content to the requested device

    Notify the user of successful delivery

**else**

    Extend the search to the regional computing infrastructure

    **if** *content is found in the regional server* **then**

        Stream the content to the requested device

**else**

        Search the attached base stations

        **if** *content is found in a base station* **then**

            Stream the content to the requested device

        **else**

            Forward the request to the Cloud Server

        **end**

    **end**

**end**

---

minimize the data forwarded to the cloud for processing. The cloud servers are tasked with storing all users' data, which is uploaded during off-peak hours.

The delay faced by the cloud is calculated as follows:

$$DC_t = D_{me} + D_{er} + D_{rc} \tag{10}$$

where $DC_t$ is the total delay faced while interacting with cloud servers, $D_{me}$ is the delay between mobile device and edge, $D_{er}$ is the delay between edge and regional servers and similarly the $D_{rc}$ is the delay between regional servers and cloud servers.

The major problem with the cloud is that traffic around the world connects to it, which increases the queuing delay ($D_q$); similarly, due to the overload, it also increases the processing delay ($D_{proc}$). Furthermore, due to the high delay, the transmitter devices consider it a packet loss and transmit another, creating an avalanche effect. Processing and storing the data nearby is necessary instead of transmitting it to distanced, congested servers to overcome the avalanche effect. In cellular communication, a delay up to 50 ms is considered good quality; however, once the delay increases from 200 ms, it has terrible effects on communication (*Kumar, Manjunath & Kuri, 2008*).

Algorithm 1 receives a content request from the users, and the optimum content location is suggested to stream the data to the target device with minimum time and cost.

This algorithm receives the search request on the failure of a search at regional servers. The content is searched on the cloud servers. If the search is successful, the data is streamed to the target devices; in the case of failure, the request is forwarded to the regional servers and the base station.

## Energy calculation

Likewise, to delay, energy consumption is also directly proportional to distance. More energy means more operational costs for data transfer. The energy consumption in the cloud environment is calculated as follows:

$$E_{\text{tran}} = \sum_{j=1}^{n} E_{\text{tran}}(j, j+1) \tag{11}$$

$E_{tran}(j, j+1)$ represents the power consumption between consecutive stages. The power consumption increases as the stages increase.

where

$$E_{tran}(j, j+1) = \frac{D_{j,j+1} \cdot P_j}{T_{j,j+1}} \tag{12}$$

where $D_{j,j+1}$ is the distance, $P_j$ is the power and $T_{j,j+1}$ is the time.

$$E_{other} = E_{pro} + E_{stor} + E_{col} + k. \tag{13}$$

Similarly, $E_{other}$ shows energy consumption in other activities such as energy consumption in processing ($E_{pro}$), storing ($E_{stor}$), and cooling the data centers ($E_{col}$) (*uz Zaman et al., 2019*).

Therefore, the total energy utilization of communication is calculated as follows:

$$E_{total} = E_{tran} + E_{other}. \tag{14}$$

Here, $E_{total}$ represents the total power usage, while $E_{tran}$ denotes the total energy usage during data transfer, encompassing wires, switches, routers, and other devices.

It is also known that,

$$Cost_{oper} \propto E. \tag{15}$$

The power consumption ($E$) is directly proportional to the operational cost ($Cost_{oper}$); consequently, higher power consumption leads to an increase in operational costs.

## EVALUATION

This article extends the Cloud Analytic simulator to implement the proposed structure Analyst (*Cloud Analyst, 2022*). It is a Java-based simulator massively utilized to simulate cloud and edge environments. The potential reason for its use for the current scenario is that it allows setting the maximum number of users, peak hour users, and off-peak hour users. The delay, response time, and cost during the simulation are calculated. Table 4 shows the simulation parameters.

**Table 4 Simulation parameters and settings.**

| Parameters | Description |
|---|---|
| Quantity of cloud servers | 1 |
| Quantity of regional servers | 1 |
| Quantity of base stations | 5 |
| Peak hours | 03:00 PM to 09:00 PM |
| Off-peak hours | 01:00 AM to 0900 AM |
| Peak hours request | 1,000 |
| Off-peak hours users | 100 |
| Workload per request | 100 kB |
| Simulation time | 1 h |
| Delay | Round trip delay |
| Cost | Communication and processing cost |
| Simulation repetition for average calculation | 10 times |

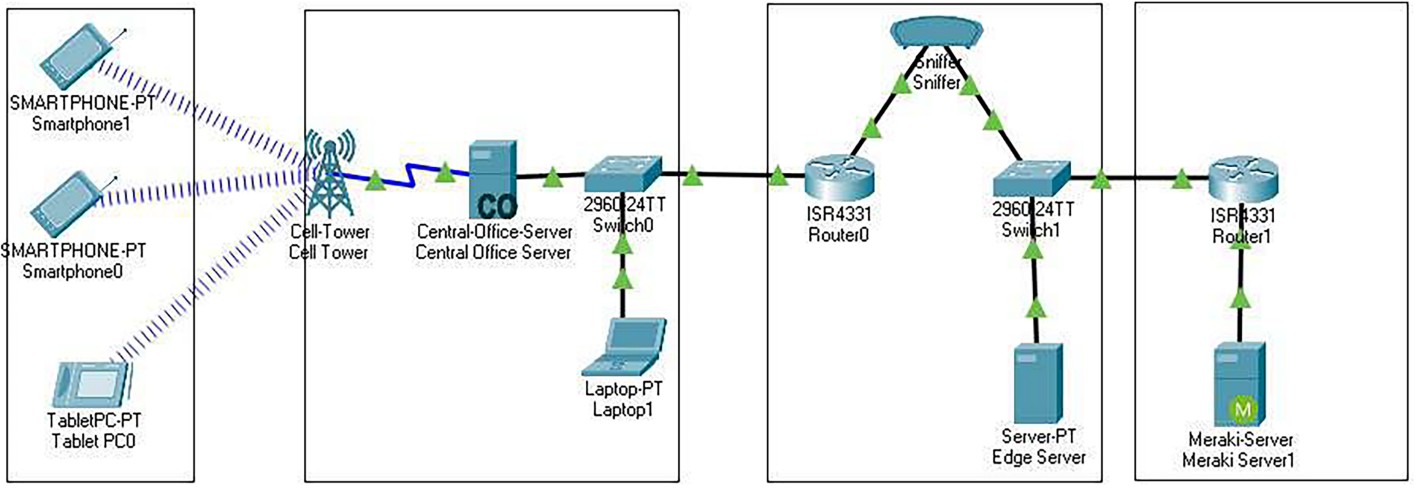

**Figure 3 Structure of the experimental setup for the proposed collaboration model.**

Similarly, we used a packet tracer (PT) simulation tool for cellular environment communication. It is developed by CISCO and widely used by engineers and scholars for network communication simulations (*Wu et al., 2017*).

## Experimental setup

The investigation is divided into three scenarios in the experimental setup, as shown in Fig. 3.

- The first scenario covers cloud servers and social media users worldwide.
- The second scenario covers the regional server, where the servers are established in a particular region, and the social media users directly access this server.
- The third scenario covers mobile edge computing, where mobile devices directly entertain the requests through the base stations.

The Cloud Analytic simulator do not support mobile-level communication. Therefore, the CISCO packet tracer is used with the Cloud Analytic to compute mobile device delay and response time.

### Cloud context

In the initial design, a single cloud server supporting 1,000 social media users worldwide was established in North America. The simulation's peak hours were from 03:00 PM to 09:00 PM, while off-peak hours spanned from 01:00 AM to 09:00 AM. During peak hours, the maximum number of users sending requests to the server was 1,000, whereas during off-peak hours, only 100 users sent requests. A workload of 100 kB was generated and uploaded from the client to the cloud server. This experiment was repeated ten times, each lasting 1 h, and the average delay and cost were calculated.

### Regional context

In the regional context, servers were established in each region to handle the regional workload. The simulation's peak hours occurred from 03:00 PM to 09:00 PM, and off-peak hours extended from 01:00 AM to 09:00 AM. During peak hours, the maximum number of users sending requests to the servers was 1,000, while only 100 users sent requests during off-peak hours. In accordance with this scenario, a workload of 100 kB was generated and executed on each regional cloud. The experiment was repeated ten times, each lasting 1 h, to calculate the average delay and cost.

### Mobile context

In the third scenario, we simulated mobile edge computing, and for MEC, we needed a simulator that may simulate the mobile cloud. CISCO packet tracer is a widely used network simulator, and after the emergence of IoT and smart systems, they integrated these devices to packet tracer (PT) (*CISCO, 2021b*). Therefore, we used a PT to simulate the mobile cloud. A base station was created, having several mobile phones. A delay was calculated by transferring the data from one mobile device to another through BS. The delay of MEC is less than a millisecond; similarly, the cost is much lower than that of Edge and Cloud servers. The 5G speed will further minimize it to be overlooked. However, MEC has several challenges, with battery capacity being the most important.

## Results analysis

The delay across the three computing paradigms shows a notable trend of increasing latency from MEC to RC and CC, as shown in Fig. 4. MEC consistently maintains the lowest delay at 12 ms, highlighting its efficient edge processing capabilities. In contrast, RC maintains a slightly higher but relatively low delay, ranging between 49 and 50 ms. However, CC exhibits the highest delay, with values fluctuating between 50 and 499 ms, signifying more variable and potentially slower processing times in the cloud. This observed progression underscores the trade-off between proximity and processing power. MEC offers the lowest latency due to its proximity to end devices, while CC, situated farther away, incurs the highest delay.

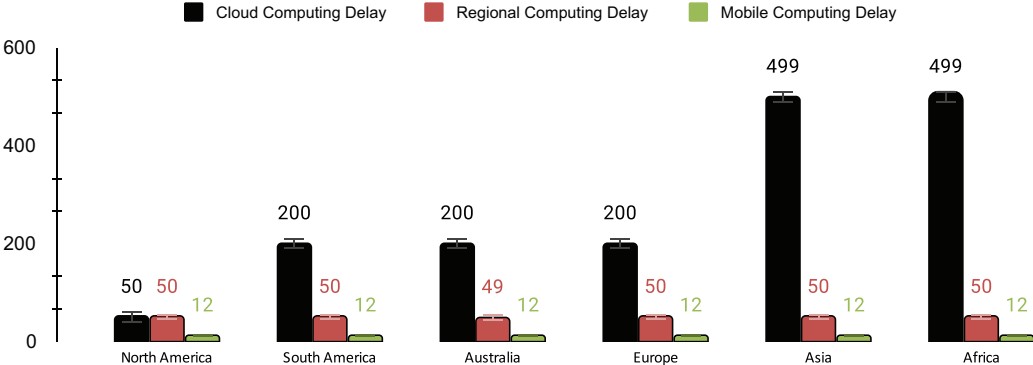

**Figure 4 Delay (millisecond) of cloud computing (CC), regional computing (RC) and mobile computing (MC).**

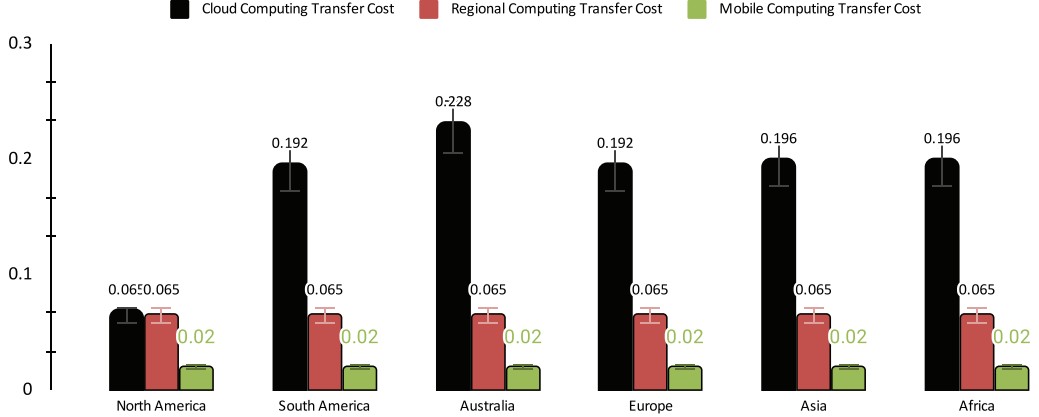

**Figure 5 Transfer cost ($) of cloud computing (CC), regional computing (RC) and mobile computing (MC).**

The cost analysis reveals distinct patterns among the three computing paradigms, as illustrated in Fig. 5. CC experiences significant variability in transfer costs, ranging from 0.065 to 0.228, with occasional spikes, indicating potential fluctuations and unpredictability in costs. In contrast, RC and MEC maintain consistent and low transfer costs, with RC at 0.065 and MEC at 0.02. This comparison underscores that, while CC may entail higher variability and costs, RC and MEC provide more stable and cost-efficient data transfer solutions. This emphasizes the advantages of leveraging regional or edge computing, particularly for applications with stringent cost considerations.

Figure 6 depicts the service level agreement (SLA) violation in milliseconds (ms) for different computing paradigms, which is calculated based on delay (*Badshah et al., 2020b*).

MEC outperforms CC and RC regarding SLA violations, providing consistently low delays and effectively meeting SLA. CC exhibits higher delay and, consequently, more SLA violations, while regional computing (RC) falls near MEC but still shows a higher delay than MEC. These results underscore the potential advantages of leveraging MEC for obtaining social big data with low latency.

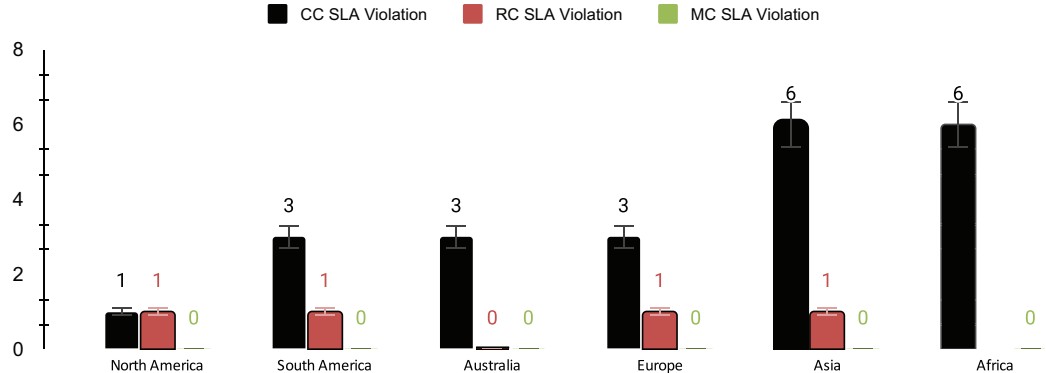

**Figure 6** SLA violations of cloud computing (CC), regional computing (RC), and mobile computing (MC).

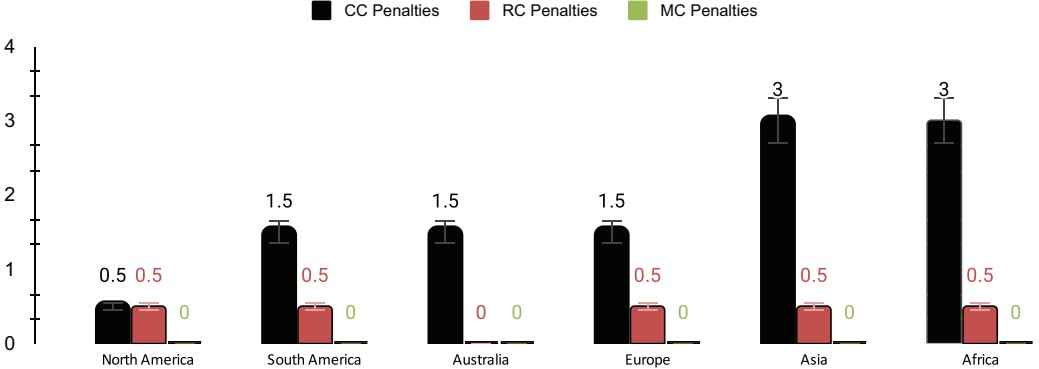

**Figure 7** Penalties ($) of cloud computing (CC), regional computing (RC), and mobile computing (MC).

The results presented in Fig. 7 indicate that cloud computing experienced penalties attributable to fluctuations in SLA violations, whereas EC consistently maintained a more stable SLA performance. MEC displayed outstanding SLA compliance, with no recorded penalties for SLA violations across all trials. These findings emphasize the considerable potential of MEC as an exceptionally efficient method for adhering to SLA without incurring extra penalties, thus reinforcing its viability for acquiring social big data.

Table 5 shows a comprehensive overview of the performance metrics for different computing paradigms. On average, cloud computing exhibits a delay of 275 ms, the highest among the paradigms, accompanied by a cost of $0.178, a 3.67% SLA violation rate, and penalties amounting to 1.83. In contrast, RC showcases a significantly lower average delay of 50 ms, a lower cost of $0.065, a minimal SLA violation rate of 0.83%, and lower penalties totaling 0.42. MEC outperforms both with an average delay of 12 ms, minimal cost ($0.02), no SLA violations, and zero penalties, highlighting its efficiency and reliability.

**Table 5 Comparison of computing paradigms. The figures are average calculations.**

| Computing | Delay ms | Cost $ | SLA violations % | Penalties |
|---|---|---|---|---|
| Cloud computing | 275 | 0.178 | 3.67 | 1.83 |
| Regional computing | 50 | 0.065 | 0.83 | 0.42 |
| Mobile computing | 12 | 0.02 | 0 | 0 |

## Comparative analysis

Numerous research endeavors have delved into the potential of edge computing to mitigate latency across diverse contexts. The article by *Kuang et al. (2021)* introduces an algorithm that harmonizes cooperative computation offloading, also resource allocation within the domain of mobile edge computing with the objective of latency reduction. Undoubtedly, edge computing brings advantages in terms of diminished delay and cost-effectiveness. However, it is crucial to underscore that a fundamental challenge lies in scalability, as edge computing inherently grapples with limitations.

Similarly, *Zarandi & Tabassum (2021)* presents a strategy to diminish latency within a multi-cell mobile edge computing network. This strategy involves optimizing offloading decisions and resource allocation. Nonetheless, this article contends with the formidable issue of scalability. Furthermore, in both *Li, Deng & Deng (2021)* and *Zeng et al. (2020)*, authors employ comparable methodologies that leverage the synergy of edge computing and mobile computing to address the reduction of communication latency and transfer costs. However, the central concern prevails—the scalability dilemmas affiliated with edge and mobile computing persist as noteworthy challenges deserving further in-depth examination.

The scenario changes when considering social big data. Here, data requirements exhibit variation. In certain instances, data necessitates processing solely at the base station level, while in others, it requires elevated processing at the cloud level. Moreover, the priority of data migration assumes significance. Scenarios demand instantaneous data transfer to the cloud, while in other cases, there is flexibility to delay data migration until off-peak hours. Notably, the proposed collaboration encompassing mobile, regional, and cloud computing addresses issues that have yet to receive adequate attention within the existing literature.

## OPEN ISSUES

While this framework has a positive impact on performance and costs, there are still open challenges with the proposed framework. Table 6 shows a detailed comparison of the technologies discussed. Mobile devices use small batteries, and powerful batteries cannot be installed; therefore, power provision is a big obstacle to mobile cloud computing. Though the battery industries increased the lifetime and capacity of batteries, they still need to be in the position to keep the mobile alive for a long time (*Shi et al., 2016*). The devices have high-quality cameras and record/stream ultra high definition videos. Running the background streaming may badly affect the performance of the mobile devices (*Avasalcai, Murturi &*

**Table 6 Comparative analysis of mobile edge computing (MEC), regional computing (RC) and cloud computing (CC) for social big data.**

| Parameters | MEC | RC | CC |
|---|---|---|---|
| Delay | Very low | Low | High |
| Response time | Very low | Low | High |
| Latency | Very low | Low | High |
| Distance from social media users | Nearer to the user | Located social media users region | Complicated network between the users and servers |
| Network model | Distributed | Distributed | Centralized |
| Security | Very few chances for security break | Fewer chances due to direct connection | High chances due to network |
| Servers mobility | Possible | Possible | Very limited |
| Real-time interaction | Real-time fast interaction | Direct connections enable real-time interaction | very limited |
| Storage capacity | Very limited | Limited storage capacity | High scalable storage capacity |

*Dustdar, 2020*). The third challenge of this proposal is that establishing data servers in different regions will need massive capital. Though these data centers will improve the market, in the initial stages, this may be resisted. Cyber rules vary from region to region. Establishing regional servers means the company will follow the exact rules. As the digital world operates globally, such rules may resist continuation in a different region.

## CONCLUSION

This article extends the concept of MEC to SMP, enabling data streaming among mobile devices rather than relying on downstream transfers from distant servers. Building on our previous work, which introduced regional computing to minimize delay and alleviate the burden on mainstream networks, this study integrates the regional computing approach with MEC. The proposed framework stores initial data on regional servers during peak hours, ensuring efficient resource utilization, and transfers it to cloud servers during off-peak hours to optimize overall performance.

## FUTURE DIRECTIONS

Initial simulations have demonstrated strong potential for this concept, highlighting the need for further exploration. To validate and refine this approach, we plan to conduct large-scale analysis through prototyping. Specifically, we intend to use Arduino WiFi four at the MEC layer and Raspberry Pi at the regional and cloud levels for implementation. Additionally, artificial intelligence will be employed to optimize offloading decisions across MEC, regional, and cloud computing layers.

### Funding

This work is supported by Taif University, Saudi Arabia, through project number (TU-DSPP-2024-67). The funders had no role in study design, data collection and analysis, decision to publish, or preparation of the manuscript.

## Grant Disclosures

The following grant information was disclosed by the authors:
Taif University, Saudi Arabia: TU-DSPP-2024-67.

## Competing Interests

The authors declare that they have no competing interests.

## Author Contributions

- Afzal Badshah conceived and designed the experiments, analyzed the data, authored or reviewed drafts of the article, and approved the final draft.
- Ameen Banjar conceived and designed the experiments, performed the experiments, authored or reviewed drafts of the article, and approved the final draft.
- Safa Habibullah performed the computation work, authored or reviewed drafts of the article, and approved the final draft.
- Abdullah Alharbi performed the experiments, prepared figures and/or tables, and approved the final draft.
- Wael Alosaimi performed the computation work, prepared figures and/or tables, and approved the final draft.
- Ali Daud conceived and designed the experiments, authored or reviewed drafts of the article, and approved the final draft.

## Data Availability

The raw data created by the simulation is available in the Supplemental File.

## Supplemental Information

Supplemental information for this article can be found online at http://dx.doi.org/10.7717/peerj-cs.2689#supplemental-information.

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
