# Peer review of "Social big data management through collaborative mobile, regional, and cloud computing"

_PeerJ Computer Science, doi:10.7717/peerj-cs.2689_

## Round 0.1 · original submission · Major Revisions

Dear authors,

Thank you for the submission. The reviewers’ comments are now available. It is not suggested that your article be published in its current format. We do, however, advise you to revise the paper in light of the reviewers’ comments and concerns before resubmitting it. The followings should also be addressed:

1. Talbe 1 and Table 5 should be corrected. "Abbereviation" should be correctly written.
2. Equations should be used with correct equation number. Many of the equations are part of the related sentences. Attention is needed for correct sentence formation.
3. Please pay special attention to the usage of abbreviations. Spell out the full term at its first mention, indicate its abbreviation in parenthesis and use the abbreviation from then on. See for example: MEC.

Best wishes,

Reviewer 1 ·

Basic reporting

Basic reporting is clear. The literature is insufficient. Self-containment is preserved.

Experimental design

The problem is unique and methodology is interesting. However, the designs need to be validated by referring them to relevant articles.

Validity of the findings

Findings are valid and results are concrete. The conclusion and future work section may be separated and enhanced.

Additional comments

For more comments pls see the attached file.

Annotated reviews are not available for download in order to protect the identity of reviewers who chose to remain anonymous.

Reviewer 2 ·

Basic reporting

English Clarity and Professionalism: The manuscript requires significant improvements in language and phrasing to ensure clarity and professionalism. For example, phrases like "degrading time-sensitive services" in the abstract can be rephrased for better readability. Consider a professional language review.

Literature References and Context: The introduction and literature review lack integration with the latest research in federated and multi-tier computing architectures. Including these references would strengthen the background and contextual relevance.

Definitions and Formal Results: Definitions for terms like "switch tasks" (page 6, line 192) and algorithms need to be explicitly stated. Additionally, formal results should include clear descriptions of parameter units and experimental conditions to ensure replicability.

Experimental design

Research Relevance: The research question is relevant and meaningful, addressing a critical gap in handling social big data using collaborative computing frameworks. However, the main objectives and contributions could be stated more concisely and categorized.

Methodological Rigor: While the study employs well-established simulation tools like Cloud Analyst and Packet Tracer, the methodology lacks sufficient detail regarding system topology and parameter units. For example, latency modeling and energy consumption parameters should specify units and provide clarity on their measurement.

Redundancy in Algorithms: Algorithms 1, 2, and 3 overlap significantly. Consolidating them into a general filtering framework could improve readability and methodological efficiency.

Replicability: The lack of detailed descriptions for key elements such as network topology, latency definitions, and experimental settings hinders the replicability of the study. Providing these details would align the methodology with high standards of rigor.

Validity of the findings

Data and Statistical Soundness: The data presented in the results section appears robust, offering a well-structured comparison of MEC, RC, and CC paradigms. However, the manuscript lacks sufficient justification for the experimental settings, including the choice of topology, the latency parameters used, and the units of measurement for the parameters. Providing this information would improve the transparency and reproducibility of the study.

Please find the further comments in the additional comments

Additional comments

1. Introduction Section:
Existing Literature: The authors should integrate references to the latest journals discussing federated and multi-tier computing architectures. These studies could offer comparisons or highlight advancements over the proposed approach. This strengthens the study's relevance and highlights its novelty.
Novelty of the Work: It would be helpful to explicitly outline how this work differs from existing literature. For instance, does it provide a novel algorithm, framework, or analysis?

2. Objectives and Contributions:
Clarity in Objectives: The stated objectives (page 3, line 67) focus on minimizing delay but do not mention content placement strategies, which are integral to the problem. Adding clarity to whether content placement strategies are considered would make the objective more robust.
Categorization of Contributions: The contributions (line 83) should be grouped into clear categories, such as methodology, theoretical contributions, and practical implications. A concise, bulleted format is recommended to make them stand out.

3. Literature Review:
Reorganization Needed: The Literature section could be reorganized to systematically compare existing solutions to the proposed one. For instance:
Edge Computing: Mention specific limitations that the proposed model overcomes.
Hybrid Computing: Highlight gaps in scalability or latency addressed by the new model.
Collaborative Frameworks: Explain why these frameworks are insufficient for handling social big data.
Citations: Errors in citations (page 3, lines 78, 79, 81) need correction and verification.

4. Methodology:
Algorithm Description: The switching tasks concept (page 6, line 192) should be clarified. For example, what specific algorithms or mechanisms are used to achieve this switch? Practical examples or pseudocode could aid understanding.
Model Clarification: In Figure 2, clarify whether the "regional computing" is part of MEC or a separate entity. The management of tasks between these layers should also be explained.
Latency and Experimental Parameters: Units for parameters such as latency and other experimental setups need to be specified clearly.

5. Technical Concerns:
Algorithm Similarities: Algorithms 1, 2, and 3 seem to follow similar patterns for filtering and forwarding data. Consolidating these into a general "filtering and forwarding algorithm" might reduce redundancy and improve readability.
Latency Modeling: On page 8, line 242, the inclusion of propagation latency for an edge at the base station requires justification.

6. Results and Analysis:
Cost Justification: The manuscript mentions that edge computing is cost-efficient compared to cloud computing. However, edge infrastructure involves higher management costs (e.g., cooling systems and maintenance). This trade-off should be addressed with supporting data or references.

7. Writing Quality and Presentation:
Language: The manuscript contains grammatical inconsistencies and awkward phrasing. For example, in the abstract, "degrading time-sensitive services" could be rephrased as "compromising time-sensitive services."
Figures and Tables: While figures and tables are generally informative, some lack adequate labeling or context. Providing a more detailed explanation for each figure or table within the text would enhance understanding.

8. Broader Impact and Challenges:
Scalability Concerns: The paper highlights limitations in edge computing scalability but does not provide concrete solutions or suggestions. Future work should address this issue in depth.
Ethical and Regulatory Implications: Establishing regional servers may raise concerns regarding data privacy and cross-border regulations. Including a brief discussion on these aspects could add depth to the study.

9. Suggestions for Future Work:
The conclusion mentions plans to expand the study. It would be beneficial to specify whether future work will focus on prototyping, simulation improvements, or real-world deployment.

---

## Round 0.2 · accepted · Accept

Dear Authors,

One of the reviewers did not respond to the invitation to evaluate the revised paper. However, the paper was accepted by another reviewer. I also conducted my own evaluation of this revised version and found it to be satisfactory. Therefore, I believe that your paper now meets the necessary standards for publication.

Kind regards,

Reviewer 1 ·

Basic reporting

The revised version is much better and clearer to read. I found no ambiguity. The literature review is improved. The article's structure is fine now. Self-containedness is preserved in the article.

Experimental design

The experimental designs and methods are unique and provide sufficient detail.

Validity of the findings

The results are concrete and findings and valid.